# Elevated Alpha-Fetoprotein in Infantile-Onset Niemann-Pick Type C Disease with Liver Involvement

**DOI:** 10.3390/children9040545

**Published:** 2022-04-12

**Authors:** Dror Kraus, Huda Abdelrahim, Orith Waisbourd-Zinman, Elena Domin, Avraham Zeharia, Orna Staretz-Chacham

**Affiliations:** 1Institute of Neurology, Schneider Children’s Medical Center of Israel, Petach-Tikva 4920235, Israel; drork@clalit.org.il (D.K.); neuro2@clalit.org.il (H.A.); 2Sackler Faculty of Medicine, Tel Aviv University, Tel Aviv 69978, Israel; oritwz@gmail.com (O.W.-Z.); azeharia@walla.com (A.Z.); 3Institute of Gastroenterology, Nutrition and Liver Diseases, Schneider Children’s Medical Center of Israel, Petach-Tikva 4920235, Israel; 4Clinical Biochemistry (Metabolic) Laboratory, Sheba Medical Center, Ramat Gan 52621, Israel; elena.dumin@sheba.health.gov.il; 5Day Hospitalization Department, Schneider Children’s Medical Center of Israel, Petach-Tikva 4920235, Israel; 6Pediatric Metabolic Clinic, Pediatric Division, Soroka Medical Center, Ben-Gurion University, Beer Sheva 8480101, Israel

**Keywords:** Niemann-Pick disease type C, Alpha-fetoprotein, hepatic involvement, cholestasis

## Abstract

Niemann-Pick disease type C (NPC) is a rare autosomal recessive neuro-visceral lipid storage disease. We describe nine cases of infantile-onset NPC with various genetic mutations in the NPC1 gene, which presented with neonatal cholestasis. Serum alpha-fetoprotein (AFP) levels were obtained as part of their workup during the first four months of life. In eight of nine (89%) patients, serum AFP demonstrated elevated levels. Seven infants displayed marked elevations, ranging from 4 to 300 times the upper limit for age-adjusted norms. In most patients, AFP levels peaked during the initial test and declined over time as cholestasis resolved. We conclude that elevated AFP levels are a common, although non-specific, marker for NPC-associated liver disease. These findings demonstrate the benefit of including AFP levels in the workup of neonatal liver disease, especially if there is accompanied cholestasis and if NPC is suspected.

## 1. Introduction

Niemann-Pick type C (NPC, MIM# 257220) is an autosomal recessive neuro-visceral lipid storage disease with an estimated prevalence of 1:120,000–150,000 live births [1,2]. It is caused by a primary defect in lysosomal cholesterol trafficking that leads to abnormal cholesterol esterification.

Four phenotypic forms of NPC have been defined according to the age at onset of neurologic symptoms:(1)Early infantile—presents with early jaundice with or without liver dysfunction and the subsequent development of progressive neurological deterioration that eventually leads to early death during the first years of life.(2)Late infantile—presents with language delay, gait impairment, hearing loss, vertical supranuclear gaze palsy and cataplexy. Later symptoms include seizures, progressive ataxia and dementia.(3)Juvenile—the most common form, which presents in mid-to-late childhood, with insidious onset of ataxia, school difficulties and loss of motor abilities, subsequently deteriorating to seizures and dementia.(4)Adult onset—presents with dementia or psychiatric symptoms [3,4].

Although it may present at any age, liver involvement is most characteristic of the early infantile onset form of the disease. Nevertheless, it affects more than 40% of patients with all forms of NPC. Common hepatic presentations include ascites, prolonged cholestasis and hepatomegaly (usually hepatosplenomegaly) [1,2].

In early infantile NPC, jaundice and hepatomegaly tend to resolve spontaneously during the first year of life (characteristically after 6–12 months of age). Onset of neurological symptoms commonly occurs during the first 2–3 years of life [3]. A subgroup of the early infantile form is associated with death during the first months of life, prior to the onset of neurological symptoms [2].

The gold standard for diagnosing NPC is a biochemical assay demonstrating impaired cholesterol esterification and positive filipin staining in cultured fibroblasts. Nevertheless, genetic mutation analysis is currently considered the most important tool in diagnosing NPC. The disease is caused by bi-allelic mutations in one of two genes: NPC1, which accounts for approximately 95% of NPC cases, is mapped to 18q and encodes a 1278 amino acid–containing protein. The NPC1 protein is localized to the late endosomal membrane and is involved in cholesterol trafficking [3,5,6]. NPC2 accounts for approximately 4% of cases and has been mapped to chromosome 14q24.3 [7].

Early diagnosis of NPC has become a focus of interest in the context of several emerging treatment options, which are in various phases in clinical trials: (1) The recombinant heat shock protein 70 co-inducer arimoclomol, which was shown to improve ataxia and myelination in a murine NPC model and humans [8]; (2) hydroxypropyl-beta cyclodextrin (HPβCD), which releases trapped cholesterol from lysosomes, thereby attenuating disease progression both in humans and in mouse models [9]; (3) N-Acetyl-L-Leucine, a leucine derivative that seems to improve ataxia and fine motor function in NPC1 mice models and patients [10]; (4) histone deacetylase inhibitors (vorinostat) were shown to correct the cholesterol storage defect in most Niemann-Pick C1 mutant cells. Although less effective in in vivo murine models, it is currently undergoing clinical trials in humans [11]; and (5) a number of clinical trials have examined gene therapy in NPC1 models. A recent placebo-controlled trial evaluated an AAV 9/3-based gene therapy injected into the left lateral ventricle and cisterna magna with significant improvement of survival and outcome in AAV-treated mice [12].

Early diagnosis is commonly a prerequisite to effective treatment. To that end, several novel biomarkers have been investigated in recent years: (1) Oxysterols, a group of cholesterol oxidation products which are the most established, accessible, and widely used biomarkers for diagnosing NPC to date. This group includes cholestane-3b,5a,6b-triol (C-triol) and, to a lesser extent, 7-ketocholesterol (7-KC). (2) Lysosphingomyelin-509 and some of its derivatives are important in differentiating NPC from sphingomyelinase deficiencies. (3) Bile acids such as 3b,5a,6b-trihydroxy-cholanoyl-glycine, which may be more specific than C-triol [1]. Importantly, oxysterols are not expected to be significantly elevated in early infantile NPC and may therefore not be as useful in diagnosing this particular subtype.

Additional emerging biomarkers used for diagnosing and monitoring response to treatment are: (1) N-palmitoyl-O-phosphocholineserine (PPCS), which is significantly elevated in the CSF of NPC1 patients, and correlates with the severity of neurological disease. Correspondingly, its reduction correlates with clinical response to treatment [13,14]; and (2) Tau protein, which is mainly expressed in neurons of the central nervous system and serves as an important biomarker for the presence and severity of neurodegenerative diseases, mainly Alzheimer’s disease [15]. Increased levels of tau protein were also identified in NPC1 mice [16], and may therefore correlate with response to treatment.

Alpha-fetoprotein (AFP) is a glycoprotein produced in the gut and liver during fetal life and serves as a fetal precursor of albumin. It is used as a tumor marker in pediatric and adult oncology and in the prenatal diagnosis of various congenital malformations, such as neural tube defects and abdominal wall defects [17].

Serum AFP levels are physiologically elevated in all newborns, particularly preterm neonates. Postnatal levels tend to decrease rapidly, reaching near-adult levels by a few months of age. Therefore, AFP levels should be interpreted based on the infant’s chronological and gestational age [18].

As serum AFP levels are often measured as part of the diagnostic workup for neonatal liver disease, we observed several cases of markedly elevated AFP levels in infants who were eventually diagnosed with NPC. We herewith report our findings and discuss their clinical significance in the workup of neonatal cholestasis.

## 2. Materials and Methods

We conducted a retrospective chart review of patients with a molecular diagnosis of NPC1 mutations in two tertiary medical centers in Israel. Patients with a genetic diagnosis of NPC and at least one measured AFP level during the first 3 months of life were included in the study. Demographic, clinical and genetic data were collected.

Concentrations of serum AFP were measured using a UniCel DXI 800 microparticle enzyme immunoassay system (Beckman Coulter, Brea, CA, USA) and carried out according to the manufacturer’s instructions. In brief, anti-AFP microparticles were incubated with the blood specimen, and an aliquot of the reaction mixture was transferred to the matrix cell. The matrix cell was washed out, and the anti-AFP conjugate was added to the mixed solution. A specimen was added to a reaction tube with mouse monoclonal anti-AFP alkaline phosphatase conjugate and paramagnetic particles coated with a second mouse monoclonal anti-AFP antibody. The AFP in the specimen bonded to the immobilized monoclonal anti-AFP on the solid phase with the monoclonal anti-AFP-alkaline-phosphatase conjugate and at the same time reacted with diverse antigenic sites on the specimen AFP. After incubation in a reaction container, materials bound by the solid phase were incarcerated in a magnetic field, whereas unbound materials were washed away. A chemiluminescent substrate was added to the reaction tube and the light generated by the reaction was directly proportional to the amount of AFP in the sample. The amount of analyte in the specimen was determined by means of a stored, multipoint calibration curve.

Alkaline phosphatase, GGt and bilirubin serum levels were measured using the Cobas C system manual (Roche Cobas 6000/8000 Operator’s Manual, Roche, Basel, Switzerland), according to International Federation of Clinical Chemistry and Laboratory Medicine Scientific Division, Committee on Reference Systems of Enzymes IFCC Generation 2.

## 3. Results

### 3.1. Clinical and Genetic Data

Nine patients (six males) matched the inclusion criteria. Their main clinical characteristics and respective NPC1 mutations are presented in Table 1. Cases one through five are children from several local Bedouin tribes with a high consanguinity rate, hence the cluster of patients homozygous to the same mutation. Overall, four of nine (44%) patients were born to consanguineous parents (see Table 1), all of which had a sibling with NPC. All patients with the homozygous R404Q mutation were previously described in the context of pulmonary involvement in early infantile NPC [19].

At the time of publication, four patients are still alive (age range 4.5 to 8 years). The eldest patient is in a chronic vegetative state. The five remaining patients in this cohort succumbed to the disease. Three patients died at 3–4 months of age, and two patients died at 5.5 and 9 years of age, respectively.

All patients presented with the infantile-onset form of NPC. At diagnosis, seven of nine (78%) presented with hepatosplenomegaly, and the two remaining patients presented with either isolated hepatomegaly or splenomegaly (Table 1). Eight of nine (89%) presented with cholestatic jaundice. Ascites and thrombocytopenia were rare, occurring in one patient each. Signs of hepatic failure (INR > 2) were evident in three of nine (33%) patients (data not shown).

### 3.2. AFP Levels

Age-adjusted serum AFP levels were available for nine infants with an eventual diagnosis of NPC. These are shown in Figure 1, compared with age-adjusted normal levels according to Lopez-Terrada et al. [18] Serum AFP levels were elevated in eight of nine (89%) patients from whom samples were obtained during the first 3 months of life (Table 2). In seven of nine (87.5%) patients, AFP levels were markedly elevated, ranging from 4 to 300 times the upper limit of age-adjusted norms (95th percentile for age in healthy infants).

AFP levels above 100,000 ng/mL (corresponding to the upper limit of normal levels up to 1 week of age [18]) were usually documented up to 3 months of age. AFP levels peaked during the initial test and gradually declined as cholestasis resolved in most patients. Three patients demonstrated gradual increases in AFP levels up to 3 months of age. One patient had a more protracted clinical course, with highly elevated AFP levels of 350,000 ng/mL at 17 months.

Patients who presented after the neonatal period had either mildly elevated or normal AFP levels. None of these patients had active liver disease when the samples were obtained (data not shown).

### 3.3. Other Laboratory Markers

Transaminase levels were mildly elevated in eight of nine (89%) patients with ALT and AST levels that did not exceed 150 and 500 U/L, respectively. One patient had peak levels of around 300 and 1100 U/L, respectively, at around 3 months of age, shortly before deteriorating. Direct hyperbilirubinemia was evident in eight of nine (89%) patients. Alkaline phosphatase levels were elevated in all patients, whereas age-adjusted GGT levels were elevated in four patients (Table 2).

None of the patients were diagnosed prenatally. In four patients, maternal serum AFP levels at 16 to 19 weeks of gestation were available as part of the routine, second-trimester screening for major congenital malformations. All were within normal limits.

## 4. Discussion

We present a series of nine patients from multiple ethnicities with genetically confirmed infantile-onset NPC disease. Eight of these patients presented with increased AFP levels, and all but one exhibited markedly increased levels of up to 300 times the age-adjusted norms. Elevated AFP levels were recorded during the first 3 months of life, correlated with the presence of cholestasis and decreased when cholestasis resolved. Patients tested after the age of 3 months and after resolution of the active liver disease had consistently normal or marginally elevated AFP levels. These cases illustrate the narrow time window during which this test may be helpful. We did not find any additional correlations between serum AFP levels and other clinical symptoms or laboratory findings.

In most instances, serum AFP levels serve as tumor markers. Nevertheless, increased AFP levels are well documented in several metabolic conditions, most notably tyrosinemia type-1. Case reports have linked elevated AFP levels to other metabolic diseases such as citrullinemia type-2 [17,20], twinkle variants [21], galactosemia, bile acid synthesis defects and gestational alloimmune liver disease (GALD, formerly termed neonatal hemochromatosis) [22]. Elevated AFP levels were also documented in another lysosomal storage disorder with hepatic involvement—a type-1 Gaucher patient, who developed a Gaucheroma [23].

AFP is produced in the yolk sac and in fetal hepatocytes. The adult liver is known to produce AFP during either regeneration or tumorigenesis. We hypothesized that the presence of AFP in metabolic diseases with hepatic involvement may reflect regenerative activity within a damaged liver [24]. The fact that elevated AFP levels corresponded to the presence of liver involvement in the current cohort further supports this notion. It is possible that elevated AFP levels will be found in other metabolic hepatopathies and possibly in other lysosomal storage disorders. Nevertheless, the current cohort represents the largest of its kind to document consistently elevated AFP levels in any metabolic condition.

Serum AFP levels may also be increased in conditions that do not affect the liver, such as ataxia telangiectasia and Finnish-type nephrotic syndrome [17,22]. Hereditary persistence of AFP is a rare autosomal dominant trait, yet affected individuals seem to demonstrate only modestly elevated AFP levels (<100 ng/mL) [25].

Maternal serum AFP levels taken as part of the routine pregnancy screen were within normal limits in all examined cases, implying that the liver disease in NPC1 probably begins postnatally. Therefore, this readily available exam cannot be used for prenatal diagnosis.

This clinical observation has two potential clinical implications: (1) It shows that elevated AFP levels are a common, although non-specific, marker for NPC-associated liver disease. (2) It emphasizes the importance of including AFP levels in the workup of neonatal liver disease. This simple and readily available test may provide clinicians an early indication to consider a diagnosis of NPC in the appropriate clinical setting. Expeditious diagnosis of NPC has become increasingly important vis-à-vis the emerging disease-modifying agents currently under development [1].

## Figures and Tables

**Figure 1 children-09-00545-f001:**
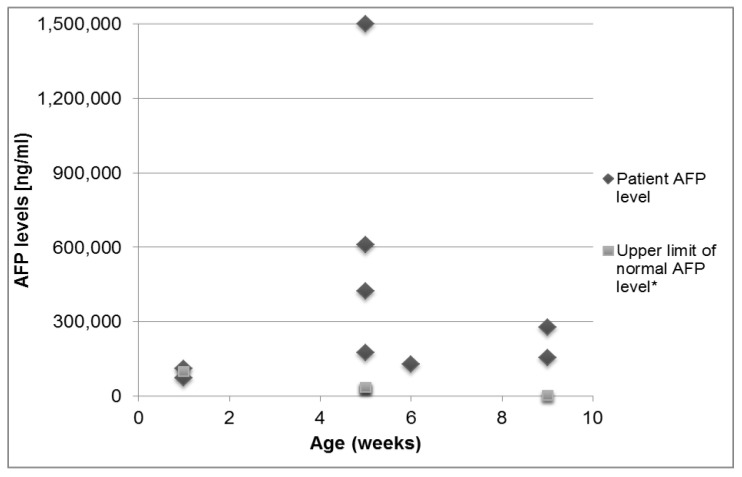
Serum AFP levels in 9 NPC patients which were tested during the first 3 months of life in comparison with age-adjusted norms. * 95th percentile of normal levels according to Lopez-Terrada et al. [18].

**Table 1 children-09-00545-t001:** Clinical and genetic characteristics of 9 patients with NPC.

Patient No.	Sex	First Manifestation	Age at Onset	Consanguinity	NPC1 Mutation
1	M	Hepatosplenomegaly	3 wk	Yes	R404Q *
2	M	Cholestasis,	1 wk	No	R404Q *
		Splenomegaly			
3	M	Cholestasis,	1 wk	No	R404Q *
		Hepatosplenomegaly			
4	F	Cholestasis,	1 mo	Yes	R404Q *
		Hepatosplenomegaly			
5	M	Cholestasis,	1 wk	Yes	R404Q *
		Hepatomegaly,			
6	F	Cholestasis,	birth	No	L1248fs/A1054T
		Hepatosplenomegaly			
7	F	Cholestasis,	1 mo	No	P166H/R1077Q
		Hepatosplenomegaly			
8	M	Cholestasis,	8 wk	Yes	F760del *
		Hepatosplenomegaly			
9	M	Cholestasis,	birth	No	F760del/S940L
		Hepatosplenomegaly			

* homozygous mutation. mo—months; wk—weeks.

**Table 2 children-09-00545-t002:** Biochemical Characteristics of 9 patients with NPC.

Patient No.	Onset of Chole-Stasis	Age at AFP Sampling	Serum AFP (ng/mL)	Upper Limit of Normal AFP *	Bili-D (mg/dL)	Bili-T (mg/dL)	ALK-P (U/L)	GGT (U/L)
1	3 wk	5 wk	**608,743**	34,672	9.4	18.5	864	41
2	1 wk	6 wk	**128,390**	879	4.89	9.63	669	398
3	1 wk	1 wk	**111,182**	98,119	16.8	28	312	102
4	1 mo	1 mo	**174,535**	34,672	2.52	4.41	450	1794
5	1 wk	1 wk	72,932	98,119	7.67	18.9	839	20
6	birth	2 mo	**277,255**	879	4.9	8.4	824	558
7	1 mo	1 mo	**422,670**	34,672	2.5	6	667	107
8	8 wk	2 mo	**155,470**	879	4.24	7.1	884	173
9	birth	1 mo	**1,495,650**	34,672	5.1	9.7	456	62

AFP—alphafeto protein; ALK-P—alkaline phosphatase; Bili-D—direct bilirubin; Bili-T—total bilirubin; GGT—gamma-glutamyltransferase; mo—months; wk—weeks. * Normal serum AFP level according to age. Bolded AFP levels represent abnormally high values.

## Data Availability

Data sharing not applicable.

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
