# Peer review of "Elevated Alpha-Fetoprotein in Infantile-Onset Niemann-Pick Type C Disease with Liver Involvement"

_children, 2022, doi:10.3390/children9040545_

Round 1

Reviewer 1 Report

The authors described that AFP levels are high in NPC infants. The clinical relevance of this non-specific marker remains unknown. 

The introduction can be improved: it is suggested to mention other NPC biomarkers described in the literature, such as oxysterols (DOI: 10.1126/scitranslmed.3001417), lysosomal volume ( 10.1172/JCI72835), and chitotriosidase (doi.org/10.1016/j.clinbiochem.2004.06.008).

The methods are described superficially: 
- it is not described how AFP and liver enzymes were measured.

Discussion:

Although it is mentioned in one sentence in the discussion, the authors could expand on why they think that AFP levels are elevated. Do the authors believe that AFP levels could be increased in other lysosomal storage disorders with liver involvement? I found one article in which a Gaucher patient with a liver Gaucherome presented elevated AFP levels (DOI: 10.1007/8904_2016_562). I think it should be mentioned in the discussion.

Reviewer 2 Report

Comments to Authors

  • Most up to date review of liver involvement in early infantile phenotype should be cited from “Seker Yilmaz B, Baruteau J, Rahim AA, Gissen P. Clinical and Molecular Features of Early Infantile Niemann Pick Type C Disease. Int J Mol Sci. 2020 Jul 17;21(14):5059.”
  • Gene therapy trials should be mentioned briefly in the introduction.
  • Have these cases been presented in any other publication? If yes, should be mentioned.
  • Age of onset should be given in Table 1.
  • Are there any other correlations between AFP and other symptoms or findings? This should be mentioned.
  • Prognosis of these patients should be given such as alive at the age of…, deceased, etc.
